# The Prevalence, Risk Factors, and Outcomes of Hepatitis E Virus Infection in Solid Organ Transplant Recipients in a Highly Endemic Area of Italy

**DOI:** 10.3390/v17040502

**Published:** 2025-03-31

**Authors:** Barbara Binda, Giovanna Picchi, Roberto Bruni, Alessandro Di Gasbarro, Elisabetta Madonna, Umbertina Villano, Giulio Pisani, Alberto Carocci, Cinzia Marcantonio, Filippo Montali, Alessandra Panarese, Francesco Pisani, Anna Rita Ciccaglione, Enea Spada

**Affiliations:** 1General and Transplant Surgery Department, San Salvatore Hospital, 67100 L’Aquila, Italy; bindabarbara@gmail.com; 2Department of Clinical Medicine, Life, Health and Environmental Sciences-MESVA, University of L’Aquila, 67100 L’Aquila, Italy; 3Infectious Diseases Department, ASL VT, PO Ospedale Belcolle Santa Rosa, 01100 Viterbo, Italy; 4Department of Infectious Diseases, Istituto Superiore di Sanita, 00161 Rome, Italy; roberto.bruni@iss.it (R.B.); elisabetta.madonna@iss.it (E.M.); umbertina.villano@iss.it (U.V.); cinzia.marcantonio@iss.it (C.M.); annarita.ciccaglione@iss.it (A.R.C.); enea.spada@iss.it (E.S.); 5Clinic of Infectious Diseases, Department of Medicine and Science of Aging, University “G. D’Annunzio” Chieti-Pescara, 66100 Chieti, Italy; reaverblade@hotmail.it; 6National Center for the Control and Evaluation of Medicines, Istituto Superiore di Sanita, 00161 Rome, Italy; giulio.pisani@iss.it (G.P.); alberto.carocci@iss.it (A.C.); 7General and Transplant Surgery Department, Dipartimento di Scienze Cliniche Applicate e Biotecnologiche-DISCAB, University of L’Aquila, 67100 L’Aquila, Italy; fmontali@ausl.pr.it (F.M.); alessandra.panarese@univaq.it (A.P.); francesco.pisani@univaq.it (F.P.)

**Keywords:** HEV, HEV–host interactions, HEV genotypes, antivirals

## Abstract

Hepatitis E virus (HEV) infection can become chronic in immunocompromised patients, like solid organ transplant recipients (SOTRs). We evaluated HEV prevalence, risk factors, and outcomes among SOTRs in a hyperendemic HEV area. Three hundred SOTRs were enrolled from April to July 2019 and tested for anti-HEV IgM and IgG and HEV RNA. Sixty-three recipients (21%) were positive for any HEV marker. HEV infection was independently associated with older age and pork liver sausage consumption. Three viremic recipients harbored genotype 3e and 3f according to HEV RNA sequencing and phylogenetic analysis. Overall, 10 recipients had markers of active/recent infection (HEV RNA and/or anti-HEV IgM) and were followed up prospectively. Five of them spontaneously resolved their HEV infection. In two recipients, HEV clearance was achieved only through immunosuppression reduction, while three needed ribavirin therapy to achieve virologic resolution. We observed a chronic course in 30% of SOTRs with active/recent HEV infection. No association was found between tacrolimus assumption and chronicization. In conclusion, we found a high prevalence of infection among SOTRs attending a transplant center in a hyperendemic Italian HEV region. Systematic screening for all HEV markers and dietary education for infection control are needed for transplant recipients.

## 1. Introduction

Hepatitis E virus (HEV) belongs to the Hepeviridae family. The taxonomy of this family has been recently updated [1]. Members of the family are now assigned to two subfamilies, five genera, and ten species. The HEV species Paslahepevirus balayani (subfamily Orthohepevirinae; genus Paslahepevirus) includes eight genotypes (HEV-1 to HEV-8) that can infect humans and wild and domestic animals. HEV-1 and HEV-2 infect only humans. The other genotypes also infect other mammals like pigs, wild boars, and deer (HEV-3 and HEV-4) and camelids (HEV-7 and HEV-8) [2,3,4,5]

HEV-1 and HEV-2 are mainly found in low-income African, Central American, and South East Asian countries, where viral transmission is mainly fecal–oral [3,4,5,6,7]. HEV-3 has spread worldwide, whereas HEV-4 is prevalent in Asia but is also present in Europe. Usually, HEV-3 and HEV-4 are transmitted by food through the ingestion of raw or undercooked meat and organs (e.g., liver and offal) of infected animals or through direct contact with them. Food-borne transmission can also occur by consuming fecally contaminated vegetables, fruits, mollusks, and drinking water. Finally, inter-human transmission of HEV-3 and HEV-4 through the transfusion of blood products and via solid organ transplantation is also possible [4,5,6,7]. Most of these zoonotic HEV infections are asymptomatic and self-limiting but in solid organ transplant recipients (SOTRs) and other immunocompromised patients (e.g., those with hematological malignancies, HIV infection, or on autoimmune disease treatments) can become chronic (in most cases HEV-3, sometimes HEV-4 and HEV-7, and rarely HEV-8) and evolve rapidly into cirrhosis [8,9,10,11,12,13]. In these patients, reduction in immunosuppressive treatment, when possible, and antiviral therapy with ribavirin are indicated [5].

Studies conducted up to the early 2010s in Italy reported very variable data on HEV infection prevalence in the general population and blood donors [14]. In the following years, the availability of highly sensitive and accurate anti-HEV antibody assays allowed researchers to conduct some reliable studies, both at regional and national scales, which greatly contributed to clarifying the epidemiological picture of HEV infection in our country [13,14,15,16]. These studies indicated that the mean crude anti-HEV IgG prevalence among blood donors in Italy was around 8–9%. There was also considerable interregional prevalence variability, with very high prevalence levels in some regions (e.g., Abruzzo), mainly attributable to local eating habits [13,14,16].

Our research group was the first to report high HEV endemicity in the Abruzzo region by conducting a prevalence survey in February–March 2014 among 313 voluntary blood donors residing in L’Aquila, the Abruzzo regional capital [13]. The detected anti-HEV IgG prevalence was 49%, and the consumption of raw or poorly cooked pork liver sausages was the only independent predictor of HEV infection. Subsequently, we performed two different nationwide HEV prevalence surveys among blood donors [14,16]. In 2015–2016, the anti-HEV IgG prevalence figures among donors from the Abruzzo region and L’Aquila were 22.8% and 31.6%, respectively [14]; in 2017–2019, we detected rates of 30% and 40%, respectively [16]. Temporal variations in anti-HEV IgG prevalence among blood donors in the same geographical area and using the same assay have already been reported in other countries, even across a longer time span [16]. Furthermore, a prospective incidence study conducted among blood donors in L’Aquila during 2013–2014 found an incidence rate of 2.1/100 person/years [17]. Such an incidence figure is much higher than that observed in the general population and blood donors from other European countries and the United States and approached the incidence rates estimated in immunocompromised patients [17]

Despite this, thus far, few Italian studies have reported data on the prevalence of HEV infection and the persistence of viremia in immunocompromised patients and even fewer in SOTRs [18,19,20,21,22,23,24,25].

In this study, we aimed to determine HEV infection prevalence and risk factors among SOTRs attending a regional transplant center operating in a high HEV prevalence area (Abruzzo). Also, we aimed to assess the prevalence and outcomes of chronic HEV hepatitis in these patients, describing clinical aspects, laboratory features, and therapeutic approaches.

## 2. Patients and Methods

### 2.1. Study Population, Design, and Ethics

All SOTRs attending the Regional Transplant Center of Abruzzo and Molise at “San Salvatore” Hospital in L’Aquila (Abruzzo region) for post-transplant follow-up in 2019 who signed an informed consent form were eligible for participation in this study. At that time, patients were not routinely tested for HEV infection before organ transplantation. All enrolled patients provided signed informed consent and were administered a questionnaire collecting information regarding risk factors for HEV infection. The clinical and laboratory data and transplant history were collected from the outpatient records of the enrolled patients.

This study used a cross-sectional and prospective design. In the first phase, all participants were screened for all HEV infection markers: anti-HEV IgM, anti-HEV IgG, and HEV RNA. After the screening phase, patients who tested positive for IgM and/or HEV RNA underwent a virologic, biochemical, and clinical assessment, which would then be repeated every month for a year in order to assess the infection’s evolution. According to Kamar et al., chronic hepatitis E in SOTRs is defined as viremia persistence for at least 3 months from its detection [26].

The study was conducted as part of a public health effort for the active surveillance of transplanted patients at risk of chronic HEV infection in an area with high HEV circulation.

### 2.2. Serologic Testing for Anti-HEV Antibodies

Anti-HEV IgM and anti-HEV IgG were detected using the Wantai HEV IgM ELISA and Wantai HEV IgG ELISA, respectively (Beijing WANTAI Biological Pharmacy Enterprise Co., Ltd., Beijing, China), according to the manufacturer’s instructions. In order to make it easier to describe the results of this study, SOTRs reactive to any of these infection markers were considered as HEV-positive, regardless of whether the recipient had been infected in the past (anti-HEV IgG only) or had markers of recent/active HEV infection (anti-HEV IgM and/or HEV RNA-positive). The exact group of HEV-positive SOTRs has been specified in the text wherever confusion might arise.

### 2.3. Detection of HEV RNA

HEV RNA was detected as previously described [27]: RNA was extracted from 200 µL of plasma using the QIAmp MinElute Virus Spin kit (Qiagen, Hilden, Germany); then, one half extract was used as a template for Real-Time PCR using the RealStar HEV RT-PCR kit (Altona Diagnostics, Hamburg, Germany) according to the manufacturer’s instructions.

### 2.4. Sequencing of HEV RNA

HEV RNA-positive extracts were used as a template to amplify a fragment from the ORF2 region of the HEV genome using a previously described nested Reverse Transcription PCR [27,28]. Purified PCR products were sequenced using the BigDye Terminator Cycle Sequencing Kit on an automated sequencer (Thermo Fisher/Applied Biosystems, Waltham, MA, USA). The obtained final sequences (493 nt) were deposited in GenBank (accession numbers PP898067, MZ274270, and MZ274271).

### 2.5. Phylogenetic Analyses

Phylogenetic analysis was used for sequence genotyping and sub-genotyping. A sequence dataset for genotyping was built, including the newly obtained sequences and established reference genotype sequences [27,29]. As all of the sequences of the present study proved to be genotype 3 (see Results), a separate dataset for sub-genotyping was built, including the newly obtained sequences and established reference 3 sub-genotypes (3a to 3m) [29,30]. Phylogenetic analysis was carried out using a Maximum Likelihood approach in MEGA version 12. The phylogenetic tree was constructed using the best substitution model preliminarily estimated using the Model tool in MEGA. The statistical significance of the tree was evaluated through bootstrap analysis; bootstrap values > 70 were considered significant.

All virologic analyses were performed in the Department of Infectious Disease of Istituto Superiore di Sanità (ISS) in Rome, Italy.

### 2.6. Other Laboratory Analyses

All patients underwent routine blood tests, including complete liver and kidney function, whole-blood cell count, and plasmatic determination of immunosuppressive drug levels.

### 2.7. Clinical Management of HEV RNA-Positive Recipients

HEV RNA-positive recipients underwent infectious disease consultation, liver ultrasound, and a fibroscan to assess liver damage. In patients who had a standard immunological risk, a reduction in immunosuppressive therapies was attempted, according to EASL guidelines. Specifically, the dose of calcineurin inhibitors was reduced by approximately 30% in most cases. Monthly HEV viremia monitoring was then performed in these SOTRs. Patients persistently viremic despite a four-month reduction in immunosuppression underwent antiviral therapy. These patients received a 3-month course of ribavirin at a dose of 600 mg per day, modified according to renal function, the development of side effects, and HEV RNA levels during therapy. HEV-RNA was monitored every month until the test was negative and then at one, three, and six months to confirm a sustained virologic response.

### 2.8. Statistical Analysis

The statistical analysis was performed with RStudio (version 1.3.959), IDE for R software (version 4.2.3.2). Odds Ratios and 95% Confidence Intervals for positive results for any HEV marker (i.e., HEV positivity, as reported above) were computed using the Wald Test. The association between HEV infection and other categorical variables was estimated using the *p*-value and chi-squared test. Variables with a *p*-value < 0.10 were evaluated in a multivariate logistic model. Backward selection (based on the AIC) was used to perform the multivariate model. The variables sex and age were included in the model independently of their *p*-value. The relationship between HEV infection and continuous variables was evaluated using the Mann–Whitney U test.

## 3. Results

### 3.1. Patients’ Characteristics

In 2019, approximately 430 SOTRs attended the transplant center’s outpatient clinic. Of these patients, 300 agreed to participate in the study and signed an informed consent form and were enrolled between April and July 2019, during the scheduled transplantation follow-up visit. They were mostly males (65.3%) with a mean age of 57.8 (±10.3) years and, on average, 8.6 (±7.5) years post-transplant. More than half of the recipients resided in the Abruzzo region, while the others were mostly from the two bordering regions of Lazio and Molise. A few recipients resided in the Campania region. The vast majority of SOTRs had had a kidney transplant. (Table 1).

The comparison between SOTRs positive for any HEV marker and negative ones showed that those positive were significantly older; organ donations came more often from deceased donors, but this difference was only close to significance (*p* = 0.05); between the two groups there were no differences regarding sex, region of residence, presence of comorbidities, type of transplanted organ, biochemical parameters and immunosuppressive medications (Table 1).

### 3.2. HEV Infection Prevalence and Risk Factors

In total, 63 SOTRs (21%) were positive for any HEV marker: 61 were anti-HEV IgG-positive (of whom 8 were also anti-HEV IgM-positive and 3 were HEV RNA-positive); 1 was positive for both anti-HEV IgM and HEV RNA but anti-HEV IgG-negative; and 1 tested HEV RNA-positive but negative for both anti-HEV IgM and anti-HEV IgG (see also Appendix A). Thus, in total, there were nine anti-HEV IgM-positive patients and five HEV RNA-positive patients. Out of the nine anti-HEV IgM-positive SOTRs, four were simultaneously HEV RNA-positive and anti-HEV IgM-positive, while five recipients were IgG-positive and HEV RNA-negative. All anti-HEV IgM-positive patients and the one positive for only HEV RNA were considered as actively/recently infected and were followed up prospectively. Three HEV sequences were obtained (Figure 1). The 53 recipients who tested positive for anti-HEV IgG only had a past HEV infection and were not followed up.

By analyzing the responses to the socio-demographic and risk factors questionnaire through univariate and multivariate logistic analyses (Table 2), we found that the only variables independently associated with HEV positivity were age over 65 years and eating pork liver sausages.

### 3.3. Prospective Follow-Up of Patients with Evidence of Active or Recent HEV Infection

The explosion of the COVID-19 pandemic in Italy in early 2020 heavily conditioned the second phase of our study. In particular, the restrictions imposed by the lockdown prevented the SOTRs from complying with the follow-up visit schedule. Furthermore, several months elapsed between the execution of the cross-sectional survey and the start of the prospective study. This time was needed for sample collection, their shipment to Istituto Superiore di Sanità in Rome, and the execution of all virologic tests.

Table 3 shows in detail the results of the prospective virologic follow-up of the ten actively/recently infected SOTRs. It was evident that the five recipients that were anti-HEV IgM- and IgG-positive but HEV RNA-negative at the screening survey, namely, recipient 36 (a 67-year-old male), recipient 92 (a 63-year-old male), recipient 141 (a 65-year-old male), recipient 218 (a 69-year-old male), and recipient 287 (a 64-year-old male), all had a recently resolved HEV infection, as documented by their decreasing anti-HEV IgM and IgG OD values. Thus, follow-up of these patients ceased.

For recipient 133 (a 63-year-old female), resulting in anti-HEV IgM positivity with low-level viremia (the sequencing attempt failed) at the screening survey, the lack of any subsequent follow-up prevented us from ascertaining the virologic resolution until 11 months later, when new serum samples from this patient became available. At time zero, transaminases were normal, while GGT values were slightly increased. This recipient was non-compliant, and their tacrolimus dosage had already been reduced several times during standard follow-up, maintaining a very low tacrolimus trough level during any control test. We can hypothesize that viral clearance was influenced by this behavior.

Recipient 228 (a 54-year-old female) showed low-level HEV RNA (the sequencing attempt failed) and positivity for both anti-HEV IgM and IgG at the baseline survey. Her transaminases were normal with GGT values moderately increased. Before HEV screening, she had already received a low dose of a cyclosporine regimen because of the development of breast cancer and intolerance to m-TOR inhibitors. After screening, she underwent a further slight reduction in immunosuppression in consideration of the foreseeable rapid failure of the kidney transplant. She became HEV RNA-negative a month later. The patient returned to dialysis 5 months after the diagnosis of HEV infection.

Patients 58 (a 72-year-old male) and 84 (a 29-year-old female), both HEV RNA-positive at the screening survey, were still viremic 4 months later. They harbored genotype 3e and 3f, respectively (Figure 1). The strain detected in patient 58 was one of the three strains involved in an HEV outbreak in Abruzzo in 2019 [27]. After an initial and unsuccessful reduction in immunosuppression, they were both treated with ribavirin (see below).

Recipient 119 (a 56-year-old male) tested HEV RNA-positive (genotype 3f) at the screening survey (Figure 1). Despite immunosuppression reduction, he continued to have low-level viremia for 4 months, and then, due to the detection of liver fibrosis on his fibroscan, he was administered ribavirin therapy (see below).

Figure 2 summarizes all serovirologic results and the infection evolution in all HEV-infected patients. Overall, we observed a chronic course in 30% of SOTRs followed up prospectively. No association was found between a chronic course of infection and the use of tacrolimus rather than cyclosporine A as immunosuppressive treatment. Four out of the five (80%) recipients experienced a spontaneous resolution of infection and were treated with tacrolimus. A non-significant different proportion (two out of three recipients were administered tacrolimus) was found among those that developed chronic hepatitis (*p* = 1.0). Likewise, among our recipients, no association was found between a chronic course of HEV infection and low ALT/AST levels or a low platelet count upon the diagnosis of HEV infection.

### 3.4. Clinical–Laboratory Features and Outcome of Chronically Infected SOTRs

Recipient 58 (a 72-year-old male) underwent a kidney transplant from a deceased donor 9 years earlier. He had a complex medical history: chronic HCV infection, successfully treated with interferon 18 years earlier, and bilateral kidney cancer surgically removed before the transplant, in addition to skin cancer, cardiovascular diseases, and recurrent lithiasic cholangitis post cholecystectomy after transplant. His e-GFR was 53 mL/min/1.73 mq, remaining stable over time. He was on maintenance immunosuppressive therapy with a low dose of prednisone, calcineurin inhibitors (cyclosporine), and m-TOR inhibitors (everolimus) because of his cancer history. Four months after his anti-HEV-positive screening test, he presented a three-fold elevation of transaminases and GGT values. Liver ultrasound documented mild–moderate steatosis and mild hepatomegaly. The fibroscan described non-constant calculated stiffness (6.9 and 13.6 KPa). As he was considered a standard immunological risk patient, a 25% reduction in cyclosporine dose was first attempted, but given the persistence of viremia, he underwent ribavirin therapy (400 mg/day) for three months. Liver enzyme levels returned to the baseline range, and a control fibroscan showed stiffness reduction (3.6 Kpa). Viral clearance was confirmed one month and one year after the end of therapy.Recipient 84 (a 29-year-old woman) was kidney transplanted from a deceased donor 5 years earlier. She was on maintenance immunosuppressive therapy with prednisone, tacrolimus, and mycophenolate mofetil. Her e-GFR was 88 mL/min/1.73 mq, remaining stable over time. She had no previous rejection episodes or pregnancy, and she was considered a standard immunological risk patient. She was HEV-RNA-positive at the screening survey and still viremic 4 months later. She showed a two-fold liver enzyme and GGT level elevation and normal liver ultrasound and fibroscan. Tacrolimus dose reduction of approximately 30% was applied, but given the persistent viral load, 3 months later, ribavirin therapy was introduced at a dose of 600 mg per day for 3 months. During therapy, she experienced mild anemia, but a dose reduction was not necessary. Viral clearance was achieved at the end of therapy and confirmed 12 months later.Recipient 119 (a 57-year-old man) underwent a kidney transplant from a deceased donor 2 years earlier. He was on maintenance immunosuppressive therapy with prednisone, mycophenolate mofetil, and tacrolimus, and his e-GFR was 70 mL/min/1.73 mq. One year before the screening test, he had shown a two–three-fold increase in liver enzyme values, which remained unchanged over time. Liver ultrasound at that time revealed moderate steatosis, screening for HBV and HCV was negative, and HEV testing was not performed because it was not available. In April 2019, he was enrolled in the study, and the screening test was positive for HEV-RNA IgM and IgG. He complained of muscle pain in his neck and shoulders. He was considered a standard immunological risk patient, and tacrolimus dose reduction of approximately 30% was promptly applied. A fibroscan showed a stiffness of 7.2 Kpa (F1–F2 fibrosis). After 3 months, he showed persistent viral load and started a 3-month therapy with ribavirin at a dose of 600 mg per day. After 45 days and 35 days, he needed two ribavirin dose reductions (400 mg and then 200 mg per day) for symptomatic anemia. Follow-up HEV-RNA tests performed after 30 and 60 days were negative, but at the end of treatment, low-level HEV RNA was once again detected. Therefore, he underwent a further one-month course of ribavirin 400 mg per day, achieving a sustained virologic response confirmed during 1-, 3-, and 6-month follow-up tests. The short length of this second course was due to ribavirin toxicity. Liver enzyme levels returned to the baseline range after the first month of ribavirin therapy.

Two of the three chronically HEV-infected patients (recipients 58 and 119) and both patients who resolved their infection after immunosuppression modulation (recipients 133 and 228) declared joint or muscle pain with resolution after healing. All patients had normal liver enzyme levels and cholestasis indices at the end of follow-up, while eGFR remained stable in all patients with active infection, except for recipient 228, who lost the graft.

## 4. Discussion

In this study, 63 out of 300 SOTRs were positive for any HEV marker, giving an overall prevalence of 21%. The prevalence of past infection was 17.6% (53/300), while that of active/recent infection was 3.3% (10/300).

In a recent systematic review and meta-analysis of 18 studies (4557 SOTRs), the pooled estimated prevalence of HEV infection (any HEV marker) in SOTRs was about 20%, while that of acute HEV infection (anti-HEV IgM+ and/or HEV RNA+) was about 4.5%. The approximate pooled prevalence according to the different solid organs were as follows: liver 27%, kidney 15%, heart 13%, lung 5.5%, and undetermined organ 29.5% [31]. In this meta-analysis, anti-HEV IgG prevalence was significantly higher in studies that used the Wantai assay compared to studies employing other assays. The highest HEV seroprevalence was reported by studies performed among liver transplant recipients in south/south-east France (about 36–38%) and Thailand (about 56%) [32,33,34].

To the best of our knowledge, only four studies have thus far investigated HEV prevalence among SOTRs in Italy [22,23,24,25]. The study by Scotto et al. was performed in southern Italy (Apulia region) during 2012–2013 and investigated 120 kidney transplant recipients (mean age 48 years) using a non-Wantai assay, detecting an anti-HEV IgG prevalence of 3.3%. Puttini et al., during 2011–2013, investigated 118 kidney transplant recipients (mean age 51 years) in central Italy (Siena, Tuscany region) by using a non-Wantai assai and found an HEV prevalence of 10.2%. In 2017, in Turin, Piedmont region (northern Italy), Zanotto et al. analyzed 120 liver transplant recipients (mean age 51 years) using a non-Wantai assay, detecting an HEV prevalence of 8.3%. Finally, De Nicola et al., during 2010–2014, tested 79 liver transplant recipients (mean age 55 years) in Milan (Lombardy, northern Italy) using a Wantai assay and found a prevalence of 33%. Considering the characteristics and results of our study and those of the above-mentioned Italian studies, it is likely that the differences in prevalence are attributable to the antibody assay used and the type of transplanted organ, but the geographical location and the mean age of the study population may also play an important role. However, a limitation of most of these studies, including the present one, is the lack of pre-transplant HEV serological surveillance data, which prevents the possibility of assessing post-transplant incidence.

Indeed, most of the recipients enrolled in our study resided in Abruzzo, a region of Italy where very high levels of HEV incidence and prevalence have been constantly documented in blood donors [13,14,16,17]. The detected prevalence of 21% among SOTRs in this study was higher than that found among Italian blood donors during 2017–2019 (8.3%) but was significantly lower than that found among blood donors in the Abruzzo region in the same years (30%). This epidemiological picture was attributed to the widespread habit among Abruzzo residents of consuming raw or poorly cooked pork liver sausages [13,14,16,17]. In fact, in the present study, the only two independent risk factors for HEV infection among SOTRs were the consumption of raw or poorly cooked pork liver sausages and age over 65 years. A similar scenario was documented in south-west France, where among blood donors, an HEV prevalence of 52% was reported [35] and a prevalence of 38.4% was found (with an annual incidence of 3.3%) in kidney and liver transplant recipients [32].

It is well known that immunocompromised patients, particularly SOTRSs, are usually strongly and steadily advised by doctors to avoid eating any kind of raw or undercooked food. How can we then explain the high HEV prevalence found among them? Eating raw or poorly cooked pork liver sausages is a hard habit to break in certain populations (like those living in the Abruzzo region or in south-west France) and is closely linked to their traditions and history. Furthermore, we know that inter-human transmission of HEV-3 and HEV-4 through the transfusion of blood products is possible and that chronic kidney or liver disease patients, as far as immunocompromised ones are concerned, are prone to need them. However, in our study, no association was found between the transfusion of blood products and HEV infection (Table 2). In consideration of the low-to-moderate anti-HEV IgG seroprevalence in blood donors and, above all, the absolute lack of donors positive for HEV RNA in two different nationwide surveys [14,16], Italy has deemed the introduction of universal HEV RNA blood donation screening unnecessary.

One of the three HEV strains isolated in this study was involved in an outbreak in Abruzzo and Lazio in 2019 caused by multiple strains and due to the consumption of raw or undercooked pork products. The molecular data suggested that the outbreak was sustained by newly imported strains, possibly through the import of pork products or live animals from outside of Abruzzo [27].

In 3 out of 10 recipients (30%) with active/recent HEV infection, a chronic course was observed, while five patients showed a spontaneous resolution of their infection. The chronicity rate reported by us was much less than that reported by Kamar (66%) and by other authors [36,37,38]. Also, differing from Kamar, no association was found between a chronic infection course and the use of tacrolimus rather than cyclosporine A or a low platelet count or a low transaminases level upon the diagnosis of HEV infection. This is probably attributable to the small number of patients with active/recent HEV infection observed in our study compared with the number observed by Kamar (85 SOTRSs) [36].

None of our patients with viremic infection complained of gastroenteric symptoms or jaundice, while the majority (four out of five) complained of joint or muscle pain, mainly in the neck or upper arms. This suggests that extrahepatic manifestation can be predominant in chronic as well as in acute HEV infection [39]. All three patients with chronic HEV infection had altered liver enzyme levels, but this was not observed in all of the viremic infections. Thus, HEV surveillance in SOT patients should be systematic and not only guided by specific symptoms and/or laboratory test alterations.

Immunosuppressive treatment reduction of about 30% was performed as suggested by guidelines in all patients with standard immunologic risk but was effective in just one out of four patients, while another patient acquired and cleared the infection during a low-dose regimen. The decision to reduce treatment can be challenging in this type of patient but can avoid the use of antiviral therapy [40].

Ribavirin is the only antiviral therapy available against HEV infection [41]. It was mainly used in the past for anti-HCV treatment. Ribavirin is a drug with a low therapeutic index, mainly for anemia occurrence, and this can be enhanced in SOT patients, especially in those with impaired kidney function. Moreover, its efficacy can be widely affected by dose reduction. In our study, two out of three patients treated with ribavirin developed anemia and one needed a dose reduction. This same patient needed a longer course of therapy because of a viremic recurrence of infection.

## 5. Conclusions

We found a high HEV infection prevalence among SOTRSs attending a transplant center in a highly endemic HEV region in Italy. Considering the immunocompromised status of these patients and the related risk of chronicization and liver damage, all SOTRSs should be systematically tested for all HEV markers, including HEV RNA, upon transplant center admission and periodically during their post-transplant follow-up, especially if residing in hyperendemic areas. The prevention of HEV infection must be pursued, together with enhancing dietary education for these patients. Further studies are needed to assess the prevalence, clinical course, and outcome of this infection in other groups of patients with immunodeficiency beyond SOT ones, especially in hyperendemic regions.

## Figures and Tables

**Figure 1 viruses-17-00502-f001:**
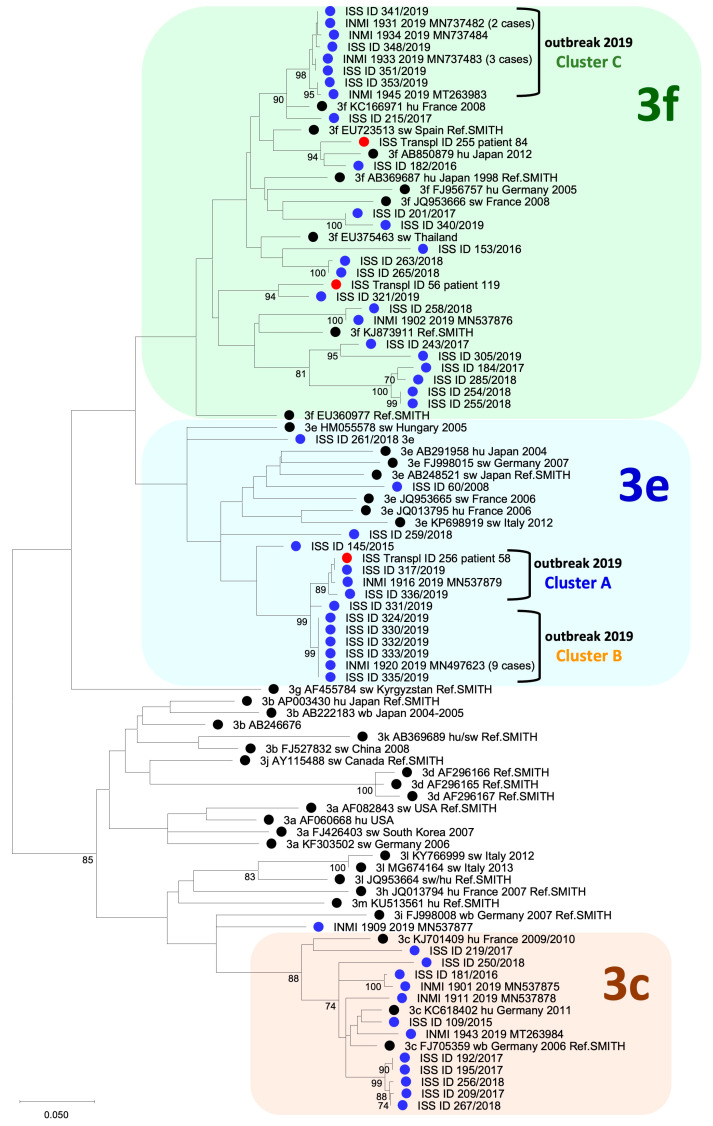
Phylogenetic tree from the analysis of the HEV sequences from the three SOTR patients (red circles) together with sequences from HEV-positive cases detected during an outbreak in Abruzzo and Lazio in 2019 (blue circles) and subtype reference sequences of HEV genotype 3 (black circles); the suffix “Ref.SMITH” in the sequence name marks the references recommended by international expert agreement [29,30]. The three molecular clusters (A, B, and C) identified in the 2019 outbreak are shown [27].

**Figure 2 viruses-17-00502-f002:**
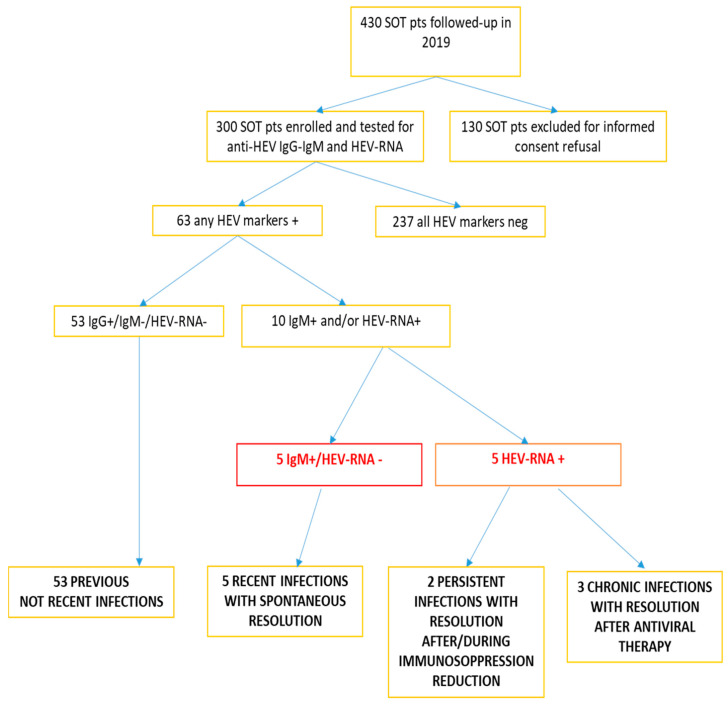
Summary of serovirological results and infection evolution in HEV-infected patients.

**Table 1 viruses-17-00502-t001:** Characteristics of the SOT recipients enrolled in the study.

	Total (300)	HEV+	HEV−	*p*-Value
**Sex**	***n*.**	***n*. (%)**	***n*. (%)**	
Female	104	18 (17.3)	86 (82.7)	0.253
Male	196	45 (23)	151 (77)	
**Age**	**(mean ± SD)**	**(mean ± SD)**	**(mean ± SD)**	
	57.8 (10.3)	62.6 (10.5)	56.5 (9.9)	0.00
**Region of residence**	***n*.**	***n*. (%)**	***n*. (%)**	
Abruzzo	171	42 (24.6)	129 (75.6)	0.081
Molise	44	10 (22.7)	34 (77.3)	0.761
Lazio	77	10 (13)	67 (87)	0.045
Campania	8	1 (12.5)	7 (87.5)	0.55
**Comorbidity**	***n*.**	***n*. (%)**	***n*. (%)**	
Yes	285	59 (20.7)	226 (79.3)	0.957
No	10	2 (20)	8 (80)	
Missing	5	2 (40)	3 (60)	-
**Transplanted organ**	***n*.**	***n*. (%)**	***n*. (%)**	
Kidney	295	60 (20.3)	235 (79.7)	0.146
Liver	4	2 (50)	2 (50)	
Kidney + liver	1	1 (100)	0 (0)	-
**Organ donor**	***n*.**	***n*. (%)**	***n*. (%)**	
Deceased	271	61 (22.5)	210 (77.5)	0.05
Living	29	2 (6.9)	27 (93.1)	
**Years from transplant**	**(mean ± SD)**	**(mean ± SD)**	**(mean ± SD)**	
	8.6 (7.5)	7.4 (6.2)	8.9 (7.8)	0.27
**Laboratory data ^1^**	**(mean ± SD)**	**(mean ± SD)**	**(mean ± SD)**	
AST (UI/L)	18.9 (7.1)	19.6 (8.7)	18.7 (6.6)	0.75
ALT (UI/L)	17.2 (10.7)	18.6 (15.4)	16.89 (9.1)	0.99
ALP (UI/L)	79.2 (34.1)	83.4 (38.8)	78.1 (32.7)	0.33
GGT (UI/L)	34.3 (40.3)	39.2 (55.7)	33.0 (35.1)	0.64
Bilirubin (mg/dL)	0.72 (0.35)	0.73 (0.31)	0.72 (0.36)	0.34
Creatinine (mg/dL)	1.63 (1.03)	1.49 (0.53)	1.67 (1.12)	0.90
Platelets (μ/nL)	216.1 (61.9)	214.5 (60.3)	216.6 (62.5)	0.85
Leukocytes (μ/nL)	6.9 (2.02)	6.7 (2.01)	7.0 (2.03)	0.45
Lymphocyte (μ/nL)	1.77 (0.73)	1.67 (0.68)	1.8 (0.75)	0.30
Lymphocyte (%)	26.4 (8.7)	25.7 (8.2)	26.5 (8.9)	0.49
**Immunosuppressants**	***n*.**	***n*. (%)**	***n*. (%)**	
Tacrolimus	219	50 (22.8)	169 (77.2)	0.2
Cyclosporine A	65	12 (18.5)	53 (81.5)	0.56
MPA/MMF	246	48 (19.5)	198 (80.5)	0.17
Methylprednisolone	272	57 (21)	215 (79)	0.95

^1^ ALT, alanine transaminase; AST, aspartate transaminase; ALP, alkaline phosphatase; GGT, gamma-glutamyl transferase; MPA, mycophenolic acid; MMF, mycophenolate mofetil.

**Table 2 viruses-17-00502-t002:** Univariate and multivariate analyses of sociodemographics and risk factors ^a^ associated with hepatitis E virus infection in solid organ transplant recipients.

					Univariate Analysis			Multivariate Analysis		
		N. Tested	HEV+ (N)	HEV+ (%)	OR	95% CI	*p*	AdjOR	95% CI	*p*
**Sex**	Female	104	18	17.3	1	-	-			
	Male	196	45	22.9	1.42	0.77–2.61	0.253	1.300	0.73–2.31	0.370
**Age** (yrs)	21–54	99	10	10.1	1	-	-			
	55–64	124	21	16.9	1.81	0.81–4.06	0.143	**2.55**	**1.32–4.94**	**0.004**
	>65	77	32	41.6	**6.33**	**2.86–14.21**	**0.000**			
**Place of residence**	Urban area	194	41	21.1	1	-	-			
	Rural area	106	22	20.7	0.97	0.55–1.75	0.936			
**Years of schooling**	0–8 yrs	140	28	20.0	1	-	-			
	≥9 yrs	157	35	22.3	1.12	0.61–1.96	0.691			
**Work with animals**	No	271	57	21.0	1	-				
	Yes	29	6	12.1	0.98	0.38–2.52	0.966			
**Swine contact**	No	279	61	21.9	1	-	-			
	Yes	17	2	11.8	0.48	0.11–2.14	0.323			
**Contact with other animals ^b^**	No	107	25	23.4	1	-	-			
	Yes	189	38	20.1	0.82	0.47–1.46	0.511			
**Hunting**	No	277	59	21.3	1					
	Yes	14	4	28.6	1.48	0.45–4.88	0.519			
**Vegetable gardening**	No	241	54	22.4	1	-	-			
	Yes	47	9	19.1	0.82	0.37–1.80	0.621			
**Eating vegetables from own or friends’ gardens**	No	120	22	18.3	1	-	-			
	Yes	174	40	23.0	1.33	0.74–2.38	0.336			
**Using manure to fertilize the garden**	No	217	44	20.3	1	-	-			
	Yes	52	13	25.0	1.31	0.64–2.66	0.454			
**Eating pork sausage ^c^**	No	126	23	18.2	-					
	Yes	164	36	21.9	1.26	0.70–2.43	0.438			
**Eating pork liver sausages ^c^**	No	240	42	17.5	1	-	-	-	-	-
	Yes	50	17	34.0	**2.43**	**1.24–4.76**	**0.008**	**2.025**	**1.11–3.68**	**0.024**
**Eating wild boar sausages ^c^**	No	264	53	20.1	1	-	-			
	Yes	26	6	23.1	1.19	0.46–3.12	0.717			
**Eating pork seasoned sausages**	No	171	29	17.0	1	-	-			
	Yes	119	30	25.2	1.651	0.93–2.93	0.086			
**Eating homemade sausages ^c^**	No	191	39	20.4	1	-	-			
	Yes	99	20	20.2	0.98	0.54–1.80	0.965			
**Eating game meat ^c^**	No	264	56	21.2	1	-	-			
	Yes	29	6	20.7	0.97	0.37–2.49	0.948			
**Eating raw seafood**	No	224	50	22.3	1	-	-			
	Yes	70	12	17.1	0.720	0.36–1.44	0.354			
**Drinking usually non-bottled water**	No	177	36	20.3	1	-	-			
	Yes	123	27	21.9	0.81	0.46–1.41	0.450			
**Blood or blood product transfusion**	No	133	25	18.8	1	-	-			
	Yes	164	38	23.2	1.30	0.74–2.29	0.359			
**Travelling abroad**	No	146	27	18.5	1	-	-			
	Yes	149	36	24.2	1.404	0.80–2.46	0.235			

^a^ Exposure to risk factors (except for demographic variables) was assessed over a lifetime; ^b^ wild boar, deer, deer, etc.; ^c^ raw or undercooked.

**Table 3 viruses-17-00502-t003:** Prospective follow-up of SOT recipients with evidence of recent and/or viremic HEV infection.

Code	Date *	IgM ^†^	IgG ^†^	HEV RNA ^†^	Date	IgM	IgG	HEV RNA	Date	IgM	IgG	HEV RNA	Date	IgM	IgG	HEV RNA	Date	IgM	IgG	HEV RNA
IN	m/y	OD	OD	cp/mL	m/y	OD	OD	cp/mL	m/y	OD	OD	copies/mL	m/y	OD	OD	cp/mL	m/y	OD	OD	cp/mL
36	4/19	**1.202**	**1.940**	NR	10/19	0.002	**1.493**	NR	FUI											
58	6/19	0.001	0.008	**10^4^–10^5^**	10/19	**2.484**	**2.319**	**10^6^**	11/19	**2.299**	**2.494**	**>10^5^**	12/19	**2.507**	**2.428**	**10^3^**	1/20	**1.980**	**2.449**	**10^3^**
84	6/19	**2.312**	**3.000**	**>10^5^**	10/19	**2.154**	**2.490**	**10^6^**	11/19	**2.475**	**2.366**	**10^4^**	12/19	**2.321**	**2.462**	**10^3^–10^4^**	1/20	**2.462**	**2.188**	NR
92	7/19	**0.523**	**2.479**	NR	10/19	**0.536**	**2.503**	NR	FUI											
119	4/19	**2.550**	**2.545**	**10^5^**	5/19	**2.517**	**2.566**	**10^5^**	6/19	**2.524**	**2.514**	**10^4^**	7/19	**2.513**	**3.000**	**>10^3^**	9/19	**2.523**	**2.490**	**10^4^**
133	4/19	**1.270**	0.001	**<10^2^**	6/19	0.001	0.001	NR	5/20	0.012	0.002	NR	9/20	0.012	0.049	NR	FUI			
141	4/19	**0.538**	**1.526**	NR	9/19	**0.483**	**1.245**	NR	FUI											
218	5/19	**0.350**	**1.746**	NR	11/19	**0.343**	**0.810**	NR	FUI											
228	5/19	**2.535**	**1.190**	**10^3^**	6/19	**2.511**	**2.525**	NR	4/20	**0.588**	**2.425**	NR	FUI							
287	5/19	**0.834**	**2.361**	NR	10/19	**0.683**	**2.115**	NR	FUI											
**Code**	**Date**	**IgM**	**IgG**	**HEV RNA**	**Date**	**IgM**	**IgG**	**HEV RNA**	**Date**	**IgM**	**IgG**	**HEV RNA**	**Date**	**IgM**	**IgG**	**HEV RNA**	**Date**	**IgM**	**IgG**	**HEV RNA**
	m/y	OD	OD	cp/mL	m/y	OD	OD	cp/mL	m/y	OD	OD	cp/mL	m/y	OD	OD	cp/mL	m/y	OD	OD	cp/mL
58	2/20	**1.575**	**2.455**	NR	4/20	**1.497**	**2.436**	NR	9/20	**0.908**	**2.350**	NR	FUI							
84	3/20	**2.467**	**2.172**	NR	4/20	**2.445**	**2.443**	NR	10/20	**2.285**	**2.372**	NR	FUI							
119	10/19	**2.484**	**2.497**	NR	11/19	**2.476**	**2.470**	NR	12/19	**2.469**	**2.481**	**<10^2^**	1/20	**2.452**	**2.457**	NR	2/20	**2.432**	**2.464**	NR
**Code**	**Date**	**IgM**	**IgG**	**HEV RNA**	**Date**	**IgM**	**IgG**	**HEV RNA**	**Date**	**IgM**	**IgG**	**HEV RNA**								
	m/y	OD	OD	cp/mL	m/y	OD	OD	cp/mL				cp/mL								
119	4/20	**2.449**	**2.456**	NR	6/20	**2.416**	**2.438**	NR	8/20	**2.328**	**2.388**	NR	FUI							

FUI, follow-up interruption; IN, identification number; IgM, anti-HEV immunoglobulin M; IgG, anti-HEV immunoglobulin G; NR, non-reactive, m/y, month/year; OD, optical density; cp/mL, copies/mL. * The first date reported in this table corresponds to that of the cross-sectional screening execution. ^†^ Positive test results are highlighted in bold.

## Data Availability

The data are available from the corresponding author upon request.

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
