# Peer review of "The Prevalence, Risk Factors, and Outcomes of Hepatitis E Virus Infection in Solid Organ Transplant Recipients in a Highly Endemic Area of Italy"

_viruses, 2025, doi:10.3390/v17040502_

Round 1

Reviewer 1 Report

Comments and Suggestions for Authors

The manuscript presents data on prevalence and course of HEV infection in solid organ transplant recipients in area of Italy where high rates of HEV seroprevalence were reported previously. The study adds new information on HEV risk factors, chronicity rates and outcomes in solid organ transplant recipients. The study is executed well. However, there are some comments that should be addressed:

  1. Please add in Introduction section more data on HEV prevalence in Abruzzo to confirm that it is area of high HEV prevalence (or even hyperendemic as stated in line 26). Usually regions with high incidence of symptomatic hepatitis E are considered as hyperendemic.
  2. Lines 103-104. “A patient reactive to any of these infection markers was considered HEV positive.” Please consider the use of term HEV positive or HEV exposed, as isolated anti-HEV IgG antibodies normally indicate the past infection. It is recommended to avoid term “HEV positive” for patients with isolated anti-HEV IgG antibodies in the context of this study, as this may mislead readers.
  3. Subsection 2.7. Please add data on HEV viremia monitoring – the time points when HEV RNA testing was performed during follow up.
  4. Table 2 containing individual patient data can be given as a supplement. Instead, please provide a table summarizing data on detection of HEV markers in patients.
  5. Lines 180-181. It is not clear if these data were obtained for all patients with HEV markers, including those with anti-HEV IgG only. Please clarify.
  6. Lines 187-188. You state that Active/recent HEV infection was a variable associated with HEV positivity. At the same time, active/recent infection was identified based on anti-HEV and HEV RNA detection, i.e. HEV positivity. The full circle. Note that data in Table 3 indicate that eating pork liver sausages was another independent risk factor, but this point is missed in the text. Please rewrite the sentence for more clarity.
  7. Table 4. Please add the column on the left indicating the patient IDs to show which data belong to a particular patient.
  8. Lines 236-237. Please remove “recent” as only patients with viremia (active infection) could developed chronic infection for obvious reason.
  9. Line 295. Please specify at which time point this patient became negative – after the cessation of the treatment or at follow-up. Was sustained virologic response assessed in this patient?
  10. Lines 353-355. Please provide information if blood donors in Italy are screened for HEV RNA.
  11. Lines 356-358. It is highly recommended to add to phylogenetic analysis HEV sequences from cited study to confirm that HEV strains from current study are close to strains that caused the outbreak. This may confirm the ongoing circulation of these strains and novelty to the study.
Comments on the Quality of English Language

Minor language editing is required.

Author Response

Comment 1: Please add in Introduction section more data on HEV prevalence in Abruzzo to confirm that it is area of high HEV prevalence (or even hyperendemic as stated in line 26). Usually regions with high incidence of symptomatic hepatitis E are considered as hyperendemic.

Response 1: To the best of our knowledge, there is no quantifiable threshold for defining the adjective “hyperendemic”.  The term “hyperendemic” expresses that the disease is constantly present in a given area at a high incidence and/or prevalence rate and affects all population age groups equally.  Apart from the definition given by the CDC (also reported by Donnelly et al. Aliment Pharmacol Ther, 2017), it does not exist any other accurate and concerted definition of HEV levels of endemicity. However, the CDC definition is difficult to apply in all settings. Besides, most importantly, it is not updated. Nowadays, the availability of highly sensitive and specific serological tests made possible a better recognition of HEV epidemiology, particularly in high income countries where geographical areas and population with very high levels of HEV prevalence have unexpectedly been found. Studies among general population and blood donors in different countries, including Europe, have shown heterogeneous (from <5% to >50%) anti-HEV IgG prevalence levels, with wide differences even in the same country. In some of these studies, geographical areas or populations characterized by particularly high IgG anti-HEV prevalence levels tending towards 50% (as well as higher ones) have been described or classified as hyperendemic. This occurred in the cases of the Midi-Pyrénées region, in southern France, and the Corsica Island (Moal et al. J Clin Microbiol. 2015; Izopet at al. J Clin Virol. 2015; Mansuy et al, Euro Surveill. 2015; Mansuy et al., Hepatology, 2016, Capai et al. Euro Surveill. 2020; Capai et al. Microorganisms 2019), the Abruzzo region in central Italy (Lucarelli et al, Euro Surveill 2016, Spada et al., Blood Transfusion 2018; Marcantonio et al. J Viral Hepat 2019; Martino et al. Eur J Public Health. 2021), the west-central Poland (Bura et al. Int J Infect Dis 2017; Bura et al., Pol J Microbiol. 2018; Capai et al. Viruses. 2019 J).

As requested, in the introduction section, we have added more data to confirm Abruzzo region as an area of high HEV prevalence: “Our research group was the first to report a high HEV endemicity in the Abruzzo region by conducting in February-March 2014 a prevalence survey among 313 voluntary blood donors residing in L’Aquila, the Abruzzo regional capital [13]. The detected anti-HEV IgG prevalence was 49%, and the consumption of raw or poor cooked pork liver sausages was the only independent predictor of HEV infection. Subsequently we performed two different nationwide HEV prevalence surveys among blood donors [14, 16]. In 2015-2016 the anti-HEV IgG prevalence figures among donors from the Abruzzo region and from L’Aquila were 22.8% and 31.6, respectively [14]; in 2017-2019 we detected respectively rates of 30% and 40% [16]. Temporal variations in anti-HEV IgG prevalence among blood donors in the same geographical area and using the same assay have already been reported in other countries, even across a longer time span [16]. Furthermore a prospective incidence study conducted among blood donors in L’Aquila during 2013-2014 found an incidence rate of 2.1/100 person/years [17]. Such incidence figure is much higher than that observed in general population and blood donors of other European countries and United States and approached the incidence rates estimated in immunocompromised patients [17].” Note that bibliographic references numeration changed a little bit.

Comment 2: Lines 103-104. “A patient reactive to any of these infection markers was considered HEV positive.” Please consider the use of term HEV positive or HEV exposed, as isolated anti-HEV IgG antibodies normally indicate the past infection. It is recommended to avoid term “HEV positive” for patients with isolated anti-HEV IgG antibodies in the context of this study, as this may mislead readers.

Response 2: We well understand the reasons for this comment, and we agree in part with the reviewer. In our opinion, at least from an epidemiological point of view, the terms ”HEV exposed” and  “HEV positive” or “HEV infected” cannot be used as synonymous terms. Exposure strictly means coming into (possible) contact with an infectious agent or source of infection, while infection (and subsequent immune response) occurs when someone is exposed and actually become infected because the infection agent passed the host’s skin-mucosal barrier, penetrated into body cells and, in the meantime, has come into contact with the immune system stimulating IgM and IgG production. So the term “exposed” can generate confusion in this context, considering the risk factors listed in our Table 3 (in the revised version Table 2), like consumption of pork liver sausages (the main risk exposure in this study). Fifty of them admitted to have eaten this food, but only 17 of them resulted actually HEV positive (i.e. became infected). Indeed, avoiding the use of the term “HEV positive” would complicate the description of the results of this study, and finally the writing of this article. Please, take in account that this term occurs many times throughout the text, being it very useful in describing the overall virological and above all the epidemiological study findings. In our epidemiological risk factors analysis, we have considered HEV positive SOTRs as a whole group, regardless they had been infected in the past or have a recent or active infection. Besides, it is very difficult to find a correct alternative term.

For these reasons we ask the reviewer to allow us the use of the term “HEV positive” in this paper . For this purpose we modified  the sentence reported in lines 103-104 as follows: “In order to make easier the description of the results in  this study, SOTRs reactive to any of these infection markers were considered as HEV positive, regardless of whether that recipient has been infected in the past (anti-HEV IgG only) or has markers of recent/active HEV infection (anti-HEV IgM and/or HEV RNA positive). The exact group of HEV positive SOTRs has been specified in the text whenever confusion might arise.We specified it also in the section 2.8 Statistical Analysis: “Odds Ratios and 95% Confidence Interval for a positive results to any HEV markers (i.e. HEV positivity, as reported above) were computed by Wald Test.”

Comment 3: Subsection 2.7. Please add data on HEV viremia monitoring – the time points when HEV RNA testing was performed during follow up.

Response 3:  We specified it in the text that: “Monthly HEV viremia monitoring was then performed in these SOTRs.” For viremic patients “HEV-RNA was monitored every month until the test was negative and then at one, three and six months to confirm a sustained virologic response”.

Indeed, as stated in the paragraph 3.3 of the results section, the restriction imposed by COVID-19 pandemic heavily conditioned the follow-up phase of our study and in particular SOTRs compliance with the follow-up visits schedule. Thus, viremia monitoring was performed whenever possible, consistent with the difficulties related to the pandemic.

Comment 4: Table 2 containing individual patient data can be given as a supplement. Instead, please provide a table summarizing data on detection of HEV markers in patients.

Response 4: We have summarized the data on detection of HEV the former Figure 2 that was modified and became Figure 1. Furthermore In the revised version of the manuscript, Table 2 has been removed and has become Supplementary Table 1.

Comment 5: Lines 180-181. It is not clear if these data were obtained for all patients with HEV markers, including those with anti-HEV IgG only. Please clarify.

Response 5.  Data reported in line 180-184  refer to SOTRs analysed in Table 1. This table showed the demographic, clinical and laboratory characteristics of the whole SOTRs study population (300 subjects) at enrollment. These 300 subjects were divided into two groups (HEV positive and HEV negative) on the basis of the HEV survey results, considering “HEV positive” as “any HEV marker positive” as specified in response 2. We specify it in the sentence  “The comparison between SOTRs positive for any HEV marker and negative ones showed that those positive were significantly older; organ donations came more often from deceased donors, but this difference was only nearly significant (p=0.05); between the two groups there were no differences regarding sex, region of residence, presence of comorbidities, type of transplanted organ, biochemical parameters and immunosuppressive medications (Table 1).”

The whole sentence has been moved in the paragraph 3.1.

Comment 6: Lines 187-188. You state that Active/recent HEV infection was a variable associated with HEV positivity. At the same time, active/recent infection was identified based on anti-HEV and HEV RNA detection, i.e. HEV positivity. The full circle. Note that data in Table 3 indicate that eating pork liver sausages was another independent risk factor, but this point is missed in the text. Please rewrite the sentence for more clarity.

Response 6: We apologize to the reviewer and rewrote the whole sentence: “By analyzing the responses to the socio-demographic and risk factors questionnaire through univariate and multivariate logistic analyses (Table 2) we found that the only variables independently associated with HEV positivity were age over 65 years and eating pork liver sausages”.

Comment 7: Table 4. Please add the column on the left indicating the patient IDs to show which data belong to a particular patient.

Response 7. We apologize again. The column with the patients’ ID in former Table 4 (now Table 3) was accidentally removed during the insertion of the tables in the text. New Table 3 (as New Table 2) has been transformed in an image to be re-oriented in the page by editors (we could not find a way to do It in Word program).

Comment 8: Lines 236-237. Please remove “recent” as only patients with viremia (active infection) could developed chronic infection for obvious reason.

Response 8: we preferred to replace the sentence “SOTRs with documented active/recent HEV infection” with the following one “SOTRs followed-up prospectively”. We preferred to do this because at the beginning of the prospective follow-up there were 5 SOTRs with active HEV infection (HEV RNA positive) and the chronicity rate would be erroneously of 60% (3/5). Instead, we think is correct to estimate the chronicity rate also considering SOTRs with serological evidences of recent post-transplant infection.

Comment 9: Line 295. Please specify at which time point this patient became negative – after the cessation of the treatment or at follow-up. Was sustained virologic response assessed in this patient?

Response 9:  We changed with the following sentence: “Therefore, recipient 119 underwent a further one-month course of ribavirin 400 mg per day, achieving a sustained virologic response confirmed at 1, 3 and 6 month follow-up tests.”

Comment 10: Lines 353-355. Please provide information if blood donors in Italy are screened for HEV RNA.

Response 10: As reported in the paper cited at reference 16 of this article, in consideration of low to moderate anti-HEV IgG seroprevalence among Italian blood donors (the overall crude and adjusted anti-HEV IgG prevalences in that survey were 8.3% and 5.5%, respectively) and above all the absolute lack of donors positive for HEV RNA (as in another previous nationwide survey) the introduction of universal HEV RNA blood donation screening in Italy does not appear justified. We added this sentence: “In consideration of low to moderate anti-HEV IgG seroprevalence in blood donors, and above all above all the absolute lack of donors positive for HEV RNA in 2 different nation-wide surveys [14, 16], Italy has deemed to be not necessary the introduction of universal HEV RNA blood donation screening.”

Comment 11: Lines 356-358. It is highly recommended to add to phylogenetic analysis HEV sequences from cited study to confirm that HEV strains from current study are close to strains that caused the outbreak. This may confirm the ongoing circulation of these strains and novelty to the study.

Response 11: The HEV sequences from the cited outbreak study were included in the dataset and the phylogenetic analysis was repeated. The previous phylogenetic tree was replaced by the newly obtained tree (former Figure 1, new Figure 2). One of the SOTR harboured one of the strains responsible for the outbreak.” The sentence at lines 356-358 was modified as follows: “One of the three HEV strains isolated in this study was involved in an outbreak in Abruzzo and Lazio in 2019 caused by multiple strains and due to consumption of raw or undercooked pork products. The molecular data suggested the outbreak was sustained by newly imported strains, possibly through the import of pork products or live animals from outside Abruzzo [28]. “

Comments on the Quality of English Language

Minor language editing is required.

Response: English language editing has been performed and improved

Reviewer 2 Report

Comments and Suggestions for Authors

The manuscript: "Prevalence, risk factors and outcome of HEV infection in solid organ transplant recipients in a high endemic area of Italy" has introduction that offers enough background for the readers. Materials and methods, however need some work; first, the inclusion/exclusion criteria are a bit vague, and perhaps Figure 2. is better suited for the description of the study group than the results. Secondly, the HEV PCR and sequencing method description is, I presume referring to the papers listed 26th, 27th, 28th and 29th in the literature, lines 114 and 124 in the text, but I suggest the authors to state it more clearly, perhaps naming the first author of the paper. In the results section, I find the table 2. unnecessary, or perhaps it can be placed as a supplementary. Table 3. is barely visible, therefore not very informative, it should either be divided or enlarged. The discussion part offers enough comparison with other data on the subject, and conclusion are supported by the results. When said corrections in the Materials and methods and results sections are made, I will find the manuscript suitable for publishing.

Author Response

Reviewer 2

The manuscript: "Prevalence, risk factors and outcome of HEV infection in solid organ transplant recipients in a high endemic area of Italy" has introduction that offers enough background for the readers. Materials and methods, however need some work; first,

Comment 1: the inclusion/exclusion criteria are a bit vague, and perhaps Figure 2. is better suited for the description of the study group than the results.

Response 1: The only exclusion criterion was the refuse to participate in it and to sign the informed consent. Many SOTRs attending the transplant center lived far from it and probably they may have refused for this reason. We modified in this way in the Patients and Methods Section: “All SOTRs attending the Regional Transplant Center of Abruzzo and Molise at “San Salvatore” Hospital in L’Aquila (Abruzzo region) for post-transplant follow-up in 2019 who signed the informed consent were eligible for participation at the study” and in the Results Section: “In 2019 approximately 430 SOTRs attended at the transplant center's outpatient clinic. Of these, 300 agreed to participate in the study, signed the informed consent and were enrolled between April and July 2019.” We specifyed it also in the former figure 2, now Figure 1, as follows (see the file attached)

Figure 1. Summary of serovirological results and infection evolution in HEV-infected patients.

Comment 2: HEV PCR and sequencing method description is, I presume referring to the papers listed 26th, 27th, 28th and 29th in the literature, lines 114 and 124 in the text, but I suggest the authors to state it more clearly, perhaps naming the first author of the paper.

Response 2: We apologize with the reviewer, the reference numbers were not formatted as requested by the journal, generating confusion. Now they are formatted according to the Author Instructions of Viruses, i.e. reference number(s) placed in square brackets before the punctuation. We also specified the MEGA Version (version 12). Reference citation by naming the first author of the paper is not foreseen by the Author Instructions of Viruses.     

Comment 3: table 2 unnecessary, or perhaps it can be placed as a supplementary

Response 3:  In the revised version of the manuscript Table 2 has become supplementary Table 1.

Reviewer 3 Report

Comments and Suggestions for Authors

I have read with interest this cross sectionnal study on HEV infection in solid organ transplat recipients. It is a decriptive study on a single center cohort of HEV in a high endemic region of Italy. 

I have few minor comments

  • The refernces format has to be homogenized
  • table 2 should be added as a supplemental file
  • What are the main genotypes and subtypes circulating in Abbruzo in the general population
  • Please detail, line 356, the strains responsible of an outbreak, do they match with the SOT patients ? were the SOT patients included in the study reported in refernce 27?
  • Line 188 the results presented are not concordant with table 3 (age and eating pork liver sausage)
  • Fig 2: add the HEV IgG result in the flow chart 
  •  

Author Response

I have read with interest this cross sectionnal study on HEV infection in solid organ transplat recipients. It is a decriptive study on a single center cohort of HEV in a high endemic region of Italy. I have few minor comments:

Comment 1: The references format has to be homogenized

Response 1: Reference have been homogenized

Comment 2: table 2 should be added as a supplemental file

Response 2: We have added Tabe2 as supplemental file (Supplementary Table 1)

Comment 3: What are the main genotypes and subtypes circulating in Abbruzo in the general population

Response 3:  The main genotypes/subtypes circulating in Abruzzo in the general population are 3f, 3e and 3c

Comment 4: Please detail, line 356, the strains responsible of an outbreak, do they match with the SOT patients ? were the SOT patients included in the study reported in reference 27?

Response 4:   The 2019 outbreak was caused by three strains (two subtype 3e strains and one 3f strain): one of them was detected in one SOTR patient described in the present study (i.e. the strain labelled “ISS Transplant ID 256 patient 58” in the phylogenetic tree of the former Fig.1, now revised and named Figure 2, see file attached). Two of the three SOTR patients of the present study were also included in the outbreak study in ref.27 (now 28), that included any symptomatic or asymptomatic HEV positive cases detected after 8 June 2019. The third patient was excluded from the outbreak study because the first positive sample was detected before 8 June 2019; now, the present study shows that the patient harboured a strain unrelated to the outbreak strains (“ISS Transpl ID 56 patient 119” in the phylogenetic tree of the revised  Fig.1).

Comment 5: Line 188 the results presented are not concordant with table 3 (age and eating pork liver sausage).

Response 5: We apologize to the reviewer and rewrote the whole sentence: “By analyzing the responses to the socio-demographic and risk factors questionnaire through univariate and multivariate logistic analyses (Table 2) we found that the only variables independently associated with HEV positivity were age over 65 years and eating pork liver sausages”.

Comment 6: Fig 2: add the HEV IgG result in the flow chart

Response 6: The HEV IgG results We simplify the description of all serologic and virologic results in this Figure and give detailed results in Supplementary Table 1, former Table 2.
